# **Sub-kilometer Scale Snow Depth Distribution on Sea Ice of Different Ages and Thickness**

Lanqing Huang<sup>1</sup>, Julienne Stroeve<sup>2,3,4</sup>, Thomas Newman<sup>1</sup>, Robbie Mallett<sup>5</sup>, Rosemary Willatt<sup>1</sup>, Lu Zhou<sup>6</sup>, Malin Johansson<sup>5</sup>, Carmen Nab<sup>1</sup>, and Alicia Fallows<sup>1</sup>

**Correspondence:** Lanqing Huang (lanqing.huang93@outlook.com)

Abstract. Accurately representing the snow depth (SND) distribution on sea ice is essential for sea ice thickness (SIT) retrievals, ecological studies, and climate modeling. Using co-located SND and SIT measurements from multiple Arctic and Antarctic campaigns, this study examines sub-kilometer-scale SND variability, considering both ice type and SIT, and identifies the most suitable statistical distributions to represent SND across different ice ages and thicknesses. First, we examine the statistical properties of SND and their dependence on SIT, finding a linear increase of SND with SIT for new and first-year ice, reflecting concurrent seasonal growth. The ratio between the standard deviation and the mean SND is referred to as the coefficient of variation (CV). A consistent  $CV \approx 0.50$  is observed to be independent of SIT, allowing variability to be estimated directly from the mean SND. Notably, lower-than-expected SND variability was observed over the flooded site, resulting in a reduced CV. Furthermore, we investigate four probability density functions (Normal, Log-normal, Gamma, and Skew) and find that the best-fit distribution depends on ice ages, SIT, deformation, and meteorological events such as snow fall and drift. Finally, SND correlation lengths derived from semi-variograms show a positive relation with SIT and are enhanced by snow drift events. The results reveal substantial differences in SND distributions across ice types and SIT during winter and summer, underscoring the importance of ice-condition-dependent parameterizations for representing sub-kilometer SND variability. These findings support improved parameterizations of SND variability at sub-grid scale in remote sensing and climate models.

#### 15 1 Introduction

Over sea ice, snow depth (SND) distributions are controlled by the presence of snow bedforms (e.g. dunes, ripples, or sastrugi) (Filhol and Sturm, 2015), along with wind redistribution, seasonal melting, underlying ice surface (Massom et al., 2001; Herzfeld et al., 2006), and ice ages (Warren et al., 1999; Iacozza and Barber, 1999; Trujillo et al., 2016; Shalina and Sandven, 2018; Kochanski et al., 2018; Wagner et al., 2022). A detailed understanding of SND distributions is essential for using coarse-grid scale SND estimates in biological studies, such as determining ideal habitats for ringed seals (Kelly et al., 2010;

<sup>&</sup>lt;sup>1</sup>Centre for Polar Observation and Modelling, Department of Earth Sciences, University College London, London, UK

<sup>&</sup>lt;sup>2</sup>Centre for Earth Observation Science (CEOS), University of Manitoba, Winnipeg, Canada

<sup>&</sup>lt;sup>3</sup>Alfred Wegener Institute, University of Bremen, Bremerhaven, Germany

<sup>&</sup>lt;sup>4</sup>National Snow and Ice Data Center (NSIDC), Cooperative Institute for Research in Environmental Sciences (CIRES), University of Colorado, Boulder, Colorado, USA

<sup>&</sup>lt;sup>5</sup>Department of Physics and Technology, UiT The Arctic University of Norway, Tromsø, Norway

<sup>&</sup>lt;sup>6</sup>Institute for Marine and Atmospheric Research Utrecht, Utrecht University, Utrecht, the Netherlands

30

Chambellant et al., 2012; Iacozza and Ferguson, 2014) or how much light penetrates under sea ice (Perovich, 1996; Nicolaus et al., 2012; Arndt et al., 2017). Moreover, improved understanding of sub-kilometer SND distributions is essential for matching coarse-resolution snow models (tens of kilometers) with the fine spatial resolution of freeboard measurements (hundreds of meters) used to derive sea ice thickness (SIT). To address this, SND redistribution approaches based on sigmoidal functions (Kwok and Cunningham, 2008) or piecewise functions (Kurtz et al., 2009; Petty et al., 2020) have been employed. Glissenaar et al. (2021) further demonstrated that small-scale snow variability exerts a significant influence on SIT estimates derived from laser freeboards. Furthermore, SND is a key parameter influencing both the thermodynamic and dynamic processes of sea ice. The spatial variation of SND is essential in understanding snow surface morphology and controls the onset of melt pond formation on level first-year ice (FYI) (Petrich et al., 2012).

Given the high spatial variability of snow cover, even over scales of just a few meters, it is often impractical to define modeling units small enough to assume uniform snow distribution within each unit. A common approach to address this challenge is to use larger modeling units (e.g. >1 km) while representing sub-kilometer variability through statistical distributions. The simplest case is to assume a uniform SND distribution (Veyssière et al., 2022). However, due to various meteorological and geophysical conditions, the snow cover is naturally highly uneven, thus the assumption of being distributed uniformly can not accurately represent the snow distributions within the sub-grid. As such, a distribution based on the mean depth has been deployed more recently (Stroeve et al., 2024; Heorton et al., 2025).

Numerous studies have examined SND statistics on terrestrial snow using Normal (Marchand and Killingtveit, 2005), Lognormal (Donald et al., 1995; Kuchment and Gelfan, 1996; Marchand and Killingtveit, 2004), and Gamma distributions (Kolberg and Gottschalk, 2010; Skaugen, 2007; Skaugen and Randen, 2013; Skaugen and Melvold, 2019). For snow on sea ice, Iacozza and Barber (2010) investigated SND on smooth landfast sea ice and reported that probability density functions (PDFs) evolved from a unimodal to a multimodal form with longer tails, indicating the development of deep snow accumulation. These PDFs effectively capture the variability in snow thickness and reflect changes in meteorological conditions, particularly snowfall and drifting snow events (Iacozza and Barber, 2010). Abraham et al. (2015) analyzed the effects of sub-grid-scale SND on the transmission of light and heat through sea ice, and implemented Rayleigh-distributed SND to improve the simulation of light fields in sea ice. Mallett et al. (2022) analyzed SND measurements taken in straight lines every 10 m along 500–1000 m transects across Arctic sea ice and found that the skew-normal distribution provided the best statistical fit to snow depth anomalies over multi-year ice (MYI), with the average coefficient of variation being around 0.42. The CV was then calculated for an additional 24 FYI sites by (Clemens-Sewall et al., 2024) who produced a similar mean CV value of 0.42.

SND distributions largely depends on the ice types. Iacozza and Barber (1999) employed geostatistical analysis using variograms, revealing that SND distribution is associated with sea ice ages. In particular, FYI showed periodic snow patterns best modeled by Gaussian and wave variograms, while MYI and rubble ice exhibited increasingly irregular distributions, modeled by spherical—Gaussian and single Gaussian variograms, respectively (Iacozza and Barber, 1999). Liston et al. (2020) suggested that ice age is strongly linked to SND and ice dynamics (i.e. snow water equivalent). Itkin et al. (2023) thoroughly investigated the seasonal and ice-type dependence of SND distribution, showing that the largest snow volumes accumulated by April, and that snow on level ice exhibited pronounced spatial heterogeneity characterized by snow dunes.

Besides, wind plays an essential role on shaping the SND variability and redistribution (Moon et al., 2019). Iacozza and Barber (1999) demonstrated that the geometric anisotropy in the snow distribution was consistently aligned with the prevailing wind directions, indicating the dominant role of the wind in shaping snow drift patterns. Wind influences the spatial distribution of SND across different ice types. For level ice, wind-blown snow tends to accumulate preferentially on younger ice (Clemens-Sewall et al., 2022), where it can form a 2.5–8 cm snow layer through redistribution from adjacent older ice. In addition, snow tends to be transported off level ice and deposited on ridges (Hames et al., 2022) and deformed ice (Merkouriadi et al., 2025) due to wind-driven redistribution. A case study in the western Arctic observed that snow depth over level ice decreased by 40% (Merkouriadi et al., 2025). Despite wind-driven redistribution of snow depth, Trujillo et al. (2016) examined the impact of a typical storm on SND at high spatial resolution (1–10 cm) over a 100×100 m area, revealing substantial local changes in snow thickness while the overall statistical distribution of surface elevation remained largely stable.

The aforementioned studies demonstrated that sea ice ages and deformation are key factors affect SND distribution on sea ice. Mallett et al. (2022) applied a skew-normal model to SND data from Soviet NP stations, demonstrating superior performance on MYI compared to younger ice. However, since no ice thickness or roughness data are available from these stations, they were unable to investigate the key relationships between SND distribution, SIT, and surface roughness. Using co-located SND and SIT data, this study extends prior work by incorporating SIT in addition to ice age as a determinant of SND distribution. The goal is to enhance knowledge of the spatial statistical characteristics of SND on sea ice and to determine the most suitable statistical distributions for sub-kilometer-scale SND across varying ice ages and thicknesses.

The study characterizes SND variability using data from multiple field campaigns carried out in the Arctic and Antarctic, which are introduced in Section 2. Co-located in-situ SND and SIT measurements encompass a wide range of ice types, including newly formed ice (NI), smooth FYI, smooth second-year ice (SYI), rough SYI, thin-snow-covered MYI in summer, and deformed MYI in winter. Section 4 presents SND statistics, identifies suitable PDFs for different ice conditions, and quantifies spatial variability via correlation lengths, revealing substantial ice-type-dependent differences in sub-kilometer-scale SND. The main findings are summarized in Section 5.

## 2 Data

### 80 2.1 Study area

We analyze SND distributions across different ice conditions and seasons using data from field campaigns in the Weddell Sea (2013) in the Antarctic, as well as N-ICE2015 (2015), Lincoln Sea (2017), MOSAiC (2019-2020), and Resolute Bay (2025) in the Arctic. The geolocation of the test sites is shown in Fig. 1.

## 2.2 Instruments

In all campaigns, SND measurements were carried out using Magnaprobes equipped with a data logger and GPS (Sturm and Holmgren, 2018). The device has a maximum measurement depth of 1.2 m. In the Lincoln Sea, MOSAiC, Resolute Bay,

**Figure 1.** The geolocations of the field campaigns in the (a) Arctic and (b) Antarctic. In (b), grey shading indicates ice cover on July 31, 2013 (final station). Blue areas represent ice shelves derived from the NSIDC EASE-Grid 2.0 Land–Ocean–Coastline–Ice Masks dataset (Meier and Stewart, 2023).

and Weddell Sea campaigns, the total thickness (snow + ice thickness) was sampled using a ground-based electromagnetic (EM) induction system (GEM-2, Geophex Ltd.) operating at multiple frequencies (Hunkeler et al., 2015, 2016). During the N-ICE2015 campaign, total ice thickness was measured with portable EM instruments (EM31 and EM31SH) mounted on a sledge (Itkin et al., 2023). For all data sets, the SND and total thickness measurements acquired follow the same track.

The Magnaprobe and GEM/EM data were processed with ice-drift correction. Magnaprobe measurements were co-located with GEM/EM measurements. SIT was calculated as the total thickness minus the snow depth. The co-located SND and SIT measurements at the test sites are shown in Fig. 2 and 3, respectively.

#### 2.3 In-situ measurements

The coincident SND and SIT observations from the study regions are presented in Fig. 4. We summarize the ice types, acquisition season, and the statistical descriptors including the mean, median, and standard deviation (STD) of both SND and SIT in Table 1, providing a comprehensive overview of snow and ice properties across the surveyed fields. Note that each test site comprises several sampling transects, and SND and SIT values for each transect are provided in Appendix A (Figs. A1 and A2).

**Figure 2.** Examples of transects of snow depth (SND) measurements. (a) MOSAiC Nloop on 14 November 2019. (b) MOSAiC Sloop on 26 December 2019. (c) MOSAiC Runway on 12 January 2020. (d) MOSAiC Summer on 3 July 2020. (e) Lincoln Sea 2017, Site 1. (f) N-ICE2015, 28 January 2015. (g) Resolute Bay, 4 April 2025. (h) Weddell Sea, floe 506. Note that the x and y coordinates are shown in relative values, with the bottom-left corner as the origin, to provide an overview of the spatial scale.

**Figure 3.** Examples of transects of sea ice thickness (SIT), co-located with the SND measurements shown in Fig. 2. (a) MOSAiC Nloop on 14 November 2019. (b) MOSAiC Sloop on 26 December 2019. (c) MOSAiC Runway on 12 January 2020. (d) MOSAiC Summer on 3 July 2020. (e) Lincoln Sea 2017, Site 1. (f) N-ICE2015, 28 January 2015. (g) Resolute Bay, 04 April 2025. (h) Weddell Sea, floe 506. Note that the x and y coordinates are shown in relative values, with the bottom-left corner as the origin, to provide an overview of the spatial scale.

## 100 2.3.1 Weddell Sea 2013

The Antarctic Winter Ecosystem and Climate Study (AWECS, ANT-XXIX/6), carried out aboard R/V Polarstern, surveyed the Weddell Sea between June and August 2013 (Lemke, 2014). During the campaign, SND and SIT measurements were collected

**Figure 4.** Violin plot of (a) sea ice thickness (SIT) and (b) snow depth (SND) for each study area, ranked in order of mean SIT. The red dot and black line represent the mean and median values, respectively.

**Table 1.** Summary of sea ice thickness (SIT) and snow depth (SND) across all sites, ranked in descending order of mean SIT. For each site, several sub-kilometer transects were measured with co-located SIT and SND values. The number of transects within each site are given in the last column. The mean, median, and standard deviation (std) of SIT and SND are calculated over all the transects for each site.

| Test sites      | Ice type | Month     | SIT      |            |         | SND      |            |         | No. of    |
|-----------------|----------|-----------|----------|------------|---------|----------|------------|---------|-----------|
|                 |          |           | mean (m) | median (m) | std (m) | mean (m) | median (m) | std (m) | transects |
| MOSAiC - NLoop  | SYI      | Oct - May | 2.97     | 2.51       | 1.84    | 0.27     | 0.23       | 0.14    | 24        |
| Lincoln Sea     | MYI      | Apr       | 2.59     | 2.34       | 1.14    | 0.33     | 0.30       | 0.23    | 9         |
| MOSAiC - Summer | SYI      | Jun-Jul   | 2.46     | 2.27       | 1.08    | 0.07     | 0.06       | 0.07    | 25        |
| N-ICE2015       | SYI      | Jan-Mar   | 1.58     | 1.43       | 0.79    | 0.50     | 0.49       | 0.16    | 4         |
| Weddell Sea     | MYI      | Jun-Aug   | 1.43     | 1.48       | 0.57    | 0.53     | 0.53       | 0.17    | 1         |
| MOSAiC - SLoop  | FYI      | Oct - May | 1.47     | 1.35       | 0.66    | 0.22     | 0.18       | 0.15    | 17        |
| Resolute Bay    | FYI      | Apr       | 1.20     | -          | -       | 0.11     | 0.09       | 0.06    | 2         |
| MOSAiC - Runway | NI       | Jan       | 1.16     | 1.13       | 0.18    | 0.12     | 0.09       | 0.08    | 3         |
| Weddell Sea     | FYI      | Jun-Aug   | 0.69     | 0.59       | 0.33    | 0.19     | 0.18       | 0.07    | 2         |

from three ice floes: PS81/503, PS81/506, and PS81/517. 503 and 506 were FYI floes, while 517 was a MYI floe (Wever et al., 2021; Arndt and Paul, 2018). The measurement locations and corresponding dates are shown in Fig. 1b.

For each floe, a set of coincident SND and SIT were conducted within a  $\sim 400 \times 400 \,\mathrm{m}$  area at a spacing of  $1-5 \,\mathrm{m}$ . For the FYI floes (503 and 506), the mean SIT and SND were 0.69 m and 0.19 m, respectively. For the MYI floe (517), the mean SIT and SND were 1.43 m and 0.53 m, respectively, shown in Table 1 and Fig. 4. The detailed SND and SIT observations along each transect are shown in Fig. A1 - A2.

## 2.3.2 N-ICE2015

The multidisciplinary Norwegian young ICE (N-ICE2015) expedition took place from January to June 2015 aboard the Norwegian research vessel R/V Lance in the area north of Svalbard. During the campaign, Lance served as a drifting research platform, moored to a sea-ice floe and drifting with the ice. This study uses coincident snow and ice thickness measurements from Floe 1 (28 January and 7 February, 2015) and Floe 2 (28 February and 14 March, 2015). Both floes were situated in an ice pack primarily composed of SYI (Granskog et al., 2018). The geolocation of the four transects is shown in Fig. 1a.

A total of 2130 co-located Magnaprobe and EM31 measurements were collected across the four transects, each within an about  $400\text{-}600 \,\text{m} \times 100\text{-}1000 \,\text{m}$  area at a spacing of  $\sim 3 \,\text{m}$ . Across the four transects, the mean SIT was 1.58 m covered by thick snow layer with a mean of 0.5 m, as shown in Table 1 and Fig. 4. The detailed SND and SIT observations along each transect are shown in Fig. A1 - A2. Note that during the N-ICE2015 expedition, widespread negative freeboard and snowpack flooding were observed, resulting from thick snow over relatively thin sea ice (Rösel et al., 2021; Merkouriadi et al., 2017). Specifically, flooding was observed in 3 of the 10 drill holes (Rösel et al., 2021). The snowpacks on the Weddell Sea MYI floe and during the N-ICE2015 field campaign are the thickest snowpacks in this study.

#### 2.3.3 Lincoln Sea 2017

In April 2017, co-located SND and SIT measurements were collected as part of the ESA CryoSat-2 Validation Experiment (CryoVEx) 2017 campaign. This study examines eight sites between Ellesmere Island and 86.3° N, shown in Fig. 1a. Eight of these sites were aligned along a northwest transect and sampled on April 12, while an additional site, located further northeast, was sampled on April 17. During the campaign, the sampling locations were covered by thick and deformed fast ice as well as drifting MYI. The sampled MYI exhibited an average SIT of 2.59 m and a substantial snow layer with an average SND of 0.33 m, see Table 1 and Fig. 4. The detailed SND and SIT observations along each transect are shown in Fig. A1 - A2.

The coincident SND and SIT data, recorded at a spacing of  $\sim 2\,\mathrm{m}$ , were utilized in this study Haas et al. (2017). In total, 6468 measurements were collected across nine sampling sites, comprising seven individual site transects within areas ranging from 150–850 m by 400–1000 m, and two transects slightly above the sub-kilometer scale (i.e.,  $0.9 \times 1.6\,\mathrm{km}^2$  and  $1.1 \times 1.4\,\mathrm{km}^2$  for Site 4 and 5, respectively). Note that the in-situ measurements were primarily taken over large, level ice patches, although some rubble and deformed ice were also sampled. As a result, the data may underestimate the mean SIT (Haas et al., 2017).

#### 2.3.4 MOSAiC 2019-2020

The year-round snow and ice thickness data from the Multidisciplinary Drifting Observatory for the Study of Arctic Climate (MOSAiC) expedition (Nicolaus et al., 2022) enabled us to analyze the SND distribution across various ice types throughout the winter and summer seasons.

During MOSAiC, SND and total thickness were measured weekly over transects with various ice conditions (Nicolaus et al., 2022; Itkin et al., 2023). The EM 18kHz measurments for total thickness are used. The transects included the Northern Loop (Nloop), Southern Loop (Sloop), and Runway during the winter season (October 24, 2019 – May 7, 2020), as well as the summer transects from June 17, 2020 - August 30, 2020. The geolocation of these transects is shown in Fig. 1a. The GPS coordinates from the Magnaprobe and EM measurements were converted into a local metric coordinate system, with the ship position as the origin, using FloeNavi toolbox (Hendricks, 2022), where all transects are displayed in Fig. B1.

A total of around 68000 samples were collected across the Nloop, Sloop, Runway, and Summer transects, all within a  $\sim 400\,\mathrm{m}^2$  area at a spacing of 1-3 m (Itkin et al., 2021). These transects encompass a range of ice conditions. The Nloop was located on deformed SYI, where the ice was both older and thicker compared to the other transects (Nicolaus et al., 2022; Wagner et al., 2022; Itkin et al., 2023). Over the autumn to winter season, the mean SIT was 2.97 m with mean SND values of 0.27 m. The Sloop, consisting primarily of FYI and remnant SYI, covered underlying frozen melt ponds and thus represents a younger, thinner ice section compared to the Nloop (Nicolaus et al., 2022; Wagner et al., 2022; Itkin et al., 2023). For the Sloop, the mean SIT and SND were 1.47 m and 0.22 m, respectively. The Runway transects were on a newly refrozen lead and therefore exhibited smooth and thin ice conditions (categorized as NI), with mean SIT and SND being 1.16 m and 0.12 m, respectively. The Summer transects were located on a completely different ice floe due to the need to reposition the ship after the ice floe broke up in mid-May. The floe was SYI with a mean SIT of 2.46 m, while the snow layer was very thin (mean of

0.07 m). The statistical details of SIT and SND measurements are summarized in Table 1 and Fig. 4. The detailed SND and SIT observations along each transect are shown in Fig. A1 - A2.

#### 2.3.5 Resolute Bay 2025

Recent fieldwork was conducted on landfast sea ice in Resolute Bay, Qikiqtaaluk Region, Nunavut, Canada, from 4–6 April, 2025. The study area, characterized by smooth FYI, is shown in Fig. 1a.

SND was measured at an interval of 1-4 m, with a total of 1559 samples collected along two transects, each covering approximately 300 × 500m. The two transects show similar mean SND values of 0.11 m. The transect on 6 April was observed to be smoother than the one on 4 April, see Fig A1(g). Note that no coincident SIT measurements were available for each SND sample; however, ice drilling in the area indicated an ice thickness of approximately 1.2 m.

#### 3 Methods

#### 3.1 Statistical distributions

We introduce four statistical distributions, i.e., Normal, Log-normal, Gamma, and Skew-normal (hereafter denoted as Skew) for fitting the observed SND samples. For a Normal distribution with mean  $\mu_g$  and standard deviation  $\sigma_g$ , the PDF is given as:

$$G(x;\mu_g,\sigma_g) = \frac{1}{\sqrt{2\pi\sigma_g^2}} e^{-\frac{(x-\mu_g)^2}{2\sigma_g^2}} \tag{1}$$

The PDF of the Log-normal distribution with mean  $e^{(\mu_l+{\sigma_l}^2/2)}$  and variance  $e^{2\mu_l+{\sigma_l}^2}\left(e^{{\sigma_l}^2}-1\right)$  follows (Gaddum, 1945):

$$LG(x;\mu,\sigma) = \frac{1}{x\sigma\sqrt{2\pi}}e^{-\frac{(\ln(x)-\mu)^2}{2\sigma^2}}$$
(2)

where  $\mu$  and  $\sigma$  are logarithm of location and scale, respectively.

The PDF of the Gamma distribution is defined as (Thom, 1958):

$$Ga(x;\alpha,\theta) = \frac{1}{\Gamma(\alpha)\theta^{\alpha}} x^{\alpha-1} e^{-x/\theta}$$
(3)

with  $\alpha > 0$  being the shape parameter,  $\theta > 0$  the scale parameter.  $\Gamma(\cdot)$  is the gamma function.

The PDF of the Skew-normal distribution is given as (O'Hagan and Leonard, 1976; Azzalini and Capitanio, 2002):

$$S(t;\omega,\alpha,\xi) = \frac{1}{\omega\sqrt{2\pi}} \left(1 + \operatorname{erf}\left(\frac{ax}{\sqrt{2}}\right)\right) e^{-(x^2/2)},$$
 (4) 
$$x = \frac{t - \xi}{\omega}$$

where  $\omega$  is the scaling parameter, a is the shape parameter,  $\xi$  is the location parameter and erf(·) is the error function.

The four PDFs are used to fit the observed SND. For each transect, the PDF parameters for each distribution are estimated using maximum likelihood estimation (MLE) (Myung, 2003). Considering that each transect has hundreds of samples, we

assume that MLE overfitting of the observed SND in the dataset is negligible. Figure 5a presents an example transect from MOSAiC Nloop (14 November, 2019), where the histogram of the observed SND is shown as a step-line plot, while the fitted distributions are displayed in different colors.

**Figure 5.** (a) Comparison of the Skew, Log-normal, Gamma, and Normal probability density functions (PDFs) with the histogram of measured snow depth (SND) values for a transect (Nloop, 14 November, 2019). The histogram were generated with a bin width of 5cm for display. (b)-(e) QQ-plot illustrates the goodness of fit. The root-mean-squared error (RMSE) between the fitted PDF and the observed SND measurements is calculated to quantify the goodness of fitting. The red dashed line is the 1:1 line.

The goodness of fit is assessed using a quantile-quantile (QQ) plot, which compares the quantiles of the observed SND values with those of the fitted distribution (Wilk and Gnanadesikan, 1968). If the fitted distribution accurately represents the data, the points in the QQ-plot should lie along the 1:1 line. Deviations from this line indicate discrepancies between the empirical and theoretical distributions. To quantify the goodness of fit, we calculate the root-mean-square error (RMSE) between the empirical and theoretical quantiles (Wilk and Gnanadesikan, 1968):

$$RMSE = \sqrt{\frac{1}{n} \sum_{i=1}^{n} \left(Q_{\text{obs},i} - Q_{\text{fit},i}\right)^2}$$
(5)

where  $Q_{\text{obs},i}$  and  $Q_{\text{fit},i}$  are the *i*th quantiles of the observed and fitted distributions (shown in the blue dots in Fig. 5b), respectively, and n is the total number of quantile points considered. A lower RMSE indicates a better fit.

Note that the histogram shown in Fig. 5a is only for display. In this study, the fitting performances are quantified by the RMSE from the QQ-plot. In contrast to histograms, which are sensitive to arbitrary binning choices, QQ-plots enable a direct

quantile–quantile comparison between observed and theoretical distributions. Consequently, fit metrics such as RMSE derived from QQ-plot are not affected by binning, providing a more objective basis for model evaluation.

#### 195 3.2 Geostatistics and variogram

Physical factors control spatial snow patterns from three aspects: local precipitation, wind, and topography/deformation (Moon et al., 2019). Snow correlation length is the distance over which SND remains spatially correlated; beyond this scale, measurements become effectively independent. Correlation length provides a statistical measure of spatial continuity and serves as a proxy for snow deformation, with larger correlation lengths often associated with prominent surface features such as ridges and dunes.

Variogram provides a measure of the spatial continuity of snow cover and how it varies with distance and orientation on sea ice. (Iacozza and Barber, 1999). The effective range derived from the variogram is used to calculate snow correlation lengths. A variogram can be computed by relating the semi-variance  $\gamma(h)$  to the lag distance (Sturm et al., 1995):

$$\gamma(h) = \frac{1}{2N} \sum_{i=1}^{N} (x_i - x_{i+h})^2 \tag{6}$$

where  $x_i$  and  $x_{i+h}$  are the sample values (measured SND) at location i and i+h respectively. N denotes the number of observations at a specific lag distance h.

The semi-variogram analyses employ the Matheron estimator with an exponential model and incorporate a non-zero nugget effect, using bin widths of 3 m. The non-zero nugget reflects either measurement error or spatial variability occurring at scales smaller than the 3 m bin size. Figure 6 presents the semi-variogram of SND measurements collected from MOSAiC Sloop on 14 November 2019, as an example. The fitted exponential model approaches the sill, and the effective range corresponds to the lag distance where the modeled semi-variance reaches 95% of the sill value. The sill represents the variance in SND, and is approximately equal to the square of the standard deviation. For lag distances smaller than the effective range, the semi-variance increases, reflecting sampling within spatially coherent structures. Beyond the range, however, the semi-variance levels off and approaches a constant value, indicating a loss of spatial correlation. This effective range will be referred to as the correlation length hereafter.

#### 215 4 Results

In this section, we first examine the statistical properties of the SND distribution (mean and standard deviation) and their relationship to ice type and SIT. We then fit the SND observations using statistical PDFs and identify the most suitable model across different ice types. Finally, we analyze the SND correlation length by computing semi-variograms and its relation to SIT and meteorological events.

## 220 4.1 Statistical properties of snow depth

In order to interpret the relation between SND and SIT in different ice types, we calculated the mean ( $\mu_{\rm SND}$ ) and standard deviation ( $\sigma_{\rm SND}$ ) of SND, along with the mean SIT ( $\mu_{\rm SIT}$ ) for each sub-kilometer transect within each site. In Fig. 7, each

Figure 6. Semi-variogram of snow depth (SND) measurements collected on the Sloop on November 14, 2019.

Figure 7. The relationship between the snow depth (SND) and sea ice thickness (SIT), where  $\mu_{SND}$  and  $\mu_{SIT}$  denote the mean SND and mean SIT values of each transect, respectively. The color and symbol (" $\circ$ " and "+") represent different test sites and ice types, respectively.

point represents a sub-kilometer transect from a specific test site. Figure 7 shows a clear linear relationship between  $\mu_{SND}$  and  $\mu_{SIT}$  for NI and FYI. This can be explained that younger and thinner ice forms and evolves alongside snow cover during the same seasonal cycle. In other words, NI and FYI grow primarily through thermodynamic processes, and its snow cover is closely linked to recent snowfall events. The limited surface roughness, minimal wind redistribution, and lack of significant

melt-refreeze history help to preserve a predictable linear relationship between  $\mu_{SND}$  and  $\mu_{SIT}$ . In contrast, no significant linear relationship is observed for samples from SYI and MYI. This lack of correlation can be attributed to factors such as rougher surfaces with deformed ice (as reported in Lincoln Sea (Haas et al., 2017) and MOSAiC Nloop (Nicolaus et al., 2022)), wind-driven snow redistribution in MOSAiC Nloop (Wagner et al., 2022), and melt-refreeze events for MOSAiC Summer loops.

The ratio between  $\sigma_{SND}$  and  $\mu_{SND}$  is known as the coefficient of variation (CV). A linear relationship is performed as (Brown, 1998)

$$\sigma_{\rm SND} = {\rm CV} \times \mu_{\rm SND}$$
 (7)

which suggests that  $\sigma_{\rm SND}$  can be predicted where the  $\mu_{\rm SND}$  is known.

Figure 8. Relationship between each transect's average snow depth (SND,  $\mu_{SND}$ ) and the corresponding standard deviation ( $\sigma_{SND}$ ). (a) The color represents the mean ice thickness (SIT) of each transect. (b) The color and symbol (" $\circ$ " and "+") represent different test sites and ice types, respectively.

Figure 8a shows the relationship between  $\sigma_{\rm SND}$  and  $\mu_{\rm SND}$  for all the the data sets. The results do not show a dependence of CV on SIT, suggesting a constant value of CV across different ice regimes. A linear fitting between  $\sigma_{\rm SND}$  and  $\mu_{\rm SND}$  in Fig. 8b gives CV = 0.5. The CV can be used to estimate the variability of SND from its mean value which is often known from snow modeling (Liston et al., 2020). Note that the linear correlation becomes weaker over MYI (Lincoln Sea and Weddell Sea) and SYI ice types (N-ICE2015), as the distribution becomes wider (Fig. 8b). These samples typically exhibit a thicker and rougher snow layer with  $\sigma_{\rm snow} > 0.2\,\mathrm{m}$  or  $\mu_{\rm SND} > 0.3\,\mathrm{m}$ . Most of the Lincoln Sea samples lie above the fitted line, indicating enhanced SND variability which can be linked to observed ice rafting events and significantly deformed surfaces.

In contrast, samples from N-ICE2015 and the Weddell Sea (MYI) fall below the fitted line, with a much lower CV than the others. Interestingly, these two sites experienced snow flooding (Rösel et al., 2021), likely due to the exceptionally thick snow cover (exceeding 0.5, m) over relatively thin sea ice of approximately 1.5, m thickness (Merkouriadi et al., 2017). When flooding occurs, the overlying snow is partially "eaten away," leading to a rapid reduction in the mean SND (Mallett et al., 2024). While flooding reduces the mean SND, a much lower-than-expected snow depth variability, leading to a reduced CV.

Two physical mechanisms contribute to the reduced SND variability observed at flooded sites. First, snow overburden causes the SND to decline more in response to flooding where snow is deeper. When there is more pressure on the flooded layer, the snow can turn into snow-ice more easily, causing more decline in total depth. This interpretation is supported by the observed higher brine wicking heights in denser snow, see Fig.4 in (Mallett et al., 2024). Consequently, flooding leads to a greater loss of snow in initially deeper locations, the overall SND variability. Second, flooding introduces heat at the ice surface, which is conducted upward through the snowpack. Thinner snow warms more rapidly than thicker snow, promoting sintering and strengthening of the thinner layer. This increased resistance to subsequent wind redistribution can further suppress variability in snow depth. Similar processes have been described by Clemens-Sewall et al. (2022) for snow on young and old sea ice.

## 255 4.2 Snow depth variability through distribution fitting

We now investigate (1) temporal variability based on samples from MOSAiC; (2) the dependence of fitting performance on ice ages and SIT across all test sites. Following the method introduced in Section 3.1, the observed SND values for each transect across the test sites are fitted using the four probability distributions: Normal, Log-normal, Gamma, and Skew. The corresponding RMSE values derived from QQ-plots are calculated to identify the most appropriate distribution within the sub-kilometer region (summarized in Table 2). We also calculate the skewness and kurtosis values over the transects to further interpret the fitting performance.

### 4.2.1 Time-series analyses

To ensure effective time-series analyses of the PDF fitting performances, we select the time series measurements conducted over the same geo-location loops (see Fig. B1 for geolocation). The fitting performances (i.e., RMSE values) are shown in Fig. 9-10.

The Nloop transect was taken over thick and deformed SYI thicker than 2 m. Before 19 December, the Log-normal distribution (green lines) provides the best fit to the observed SND, with lower RMSE compared to the other PDFs, see Fig. 9a. On 19 December, the performance of the Gamma and Skew improves sharply, coinciding with a notable drop in kurtosis values below 2, see Fig. 11a. Note that kurtosis describes the tailedness of a distribution; lower kurtosis indicates a flatter distribution with fewer extremes and more values spread around the mean. A possible explanation for this transition is the snow drift event reported on 19 December (Wagner et al., 2022), which could reduce the peakedness of the SND distribution and led to a more evenly redistributed snow layer.

From 19 December to 30 January, the performance of the Gamma (blue lines) and Skew (black lines) improves and become comparable RMSE values as Log-normal. To better understand the statistical characteristic of SND distribution, the skewness and kurtosis values have been calculated and shown in Fig. 11a. We observed a kurtosis < 2 (Fig. 11a) on 9 January, coincident with the point where the Skew distribution begins to outperform the Log-normal distribution. The dates on which the Log-normal performs best are marked by green bars in Fig. 11, suggesting that kurtosis > 2 can be a threshold above which the Log-normal distribution outperforms the Skew distribution. This threshold is confirmed at other test sites.

Figure 9. Root-mean-square error (RMSE) between the fitted probability density functions (PDFs) and the observed snow depth (SND) values. (a) Nloop. (b) Sloop. The mean sea ice thickness (SIT,  $\mu_{\rm SIT}$ ) and SIT range ( $\Delta_{\rm SIT} = 95\% {\rm SIT} - 5\% {\rm SIT}$ ) over each transect are calculated. The relevant meteorological events such as storm, snowfall, and snow drift are from (Wagner et al., 2022). Note that the gaps at November 14 and March 5 in (b) correspond to changes in the geolocation of the Sloop transect shown in Fig. B1.

After 30 January, the Log-normal's performance deteriorates significantly with a sharp increase in RMSE, while the Skew achieves the best fit with the lowest RMSE. Notably, during the snow drift period from 20 February to 20 March, including the strongest drift event on 24-25 February, the observed SND samples exhibit the lowest kurtosis (from  $\sim$  0.2-0.8). This observation can be explained by the fact that snow drift leads to a more evenly redistributed snow layer. A notable change in the shapes of the observed SND and SIT distributions before and after 30 January, as shown in Fig. 12, further explains the shift in the best-fitting PDF. The change in SND and SIT could be attributed to the under-surveying of deformed ice and ridges after January 30. Although there was no obvious shift in the transects' geolocations before and after this date, slight changes in sampling positions (e.g., avoiding highly deformed ice) may have contributed to the under-representation of deformed ice and ridges, as shown in Fig. B1. The proportion of samples with SIT > 6 m is 15% before and 3% after 30 January, indicating more samples from deformed ice prior to 30 January. However,  $\mu_{\rm SIT}$  remains relatively constant all the time (see orange triangle

Figure 10. Root-mean-square error (RMSE) between the fitted probability density functions (PDFs) and the observed snow depth (SND) values. (a) MOSAiC Runway. (b) MOSAiC Summer. The mean sea ice thickness (SIT,  $\mu_{\rm SIT}$ ) and SIT range ( $\Delta_{\rm SIT} = 95\% {\rm SIT} - 5\% {\rm SIT}$ ) over each transect are calculated.

symbols in Fig. 9a), suggesting that the deformed features cannot be adequately represented by a single mean ice thickness value. To better capture the deformation levels, we define the SIT range ( $\Delta_{\rm SIT}$ ) as the difference between the 95th and 5th percentiles of ice thickness for each transect. As shown in Fig. 9a, SIT range decreases substantially from an average value of  $6.79\,\mathrm{m}$  to  $4.13\,\mathrm{m}$  before and after 30 January, coinciding with the change in the best-fitting distribution. These results suggest that ice deformation influences SND distributions and that the SIT range metric serves as a useful parameter for identifying the most appropriate SND distribution.

For MOSAiC Sloop, where FYI dominates, a continuous increase in SIT over time indicates seasonal growth of sea ice in Fig. 9b. Before 20 February, when  $\mu_{\rm SIT}$  reaches approximately 1.5 m, the Log-normal performs best, exhibiting the lowest RMSE values. Similar to Nloop, a significant improvement in Gamma and Skew performance is observed on 26 December, after a documented snow drift event that occurred on 19 December (Wagner et al., 2022). After 20 February,  $\mu_{\rm SIT}$  is between  $1.5-2\,\mathrm{m}$ , the Gamma and Skew begin to outperform, with Gamma slightly better than Skew. From Fig. 11b, kurtosis decreases

**Figure 11.** Skewness and kurtosis values calculated from the observed snow depth (SND). The dates on which the Log-normal performs best are marked by green bars. The relevant meteorological events such as storm, snowfall, and snow drift are from (Wagner et al., 2022) are shown in (a)-(b).

below the threshold of 2 since 20 February, consistent with the transition of the best-fit PDF from Log-normal to Gamma or Skew distribution. The deterioration in Log-normal performance after 20 February is also likely driven by the sustained snow

**Figure 12.** The histograms of the observed (a) snow depth (SND) and (b) sea ice thickness (SIT) over MOSAiC Nloops. Note that 5 cm bins for SND distribution and 0.5 m bins for SIT distribution are used for displaying histograms.

drift period beginning that day and the intense drift event on 24-25 February, further supporting the interpretation that the snow drifting event leads to more evenly redistributed snow layers as observed in MOSAiC Nloop results. When  $\mu_{\rm SIT}$  reaches 2m on 7 May, the Skew gives the best fit with the smallest RMSE. The transition of the best PDF from Log-normal to Skew indicates a clear dependence of the SND distribution on SIT for FYI during the growth season, which will be further discussed in Section 4.2.2.

The MOSAiC Runway site is characterized by newly formed smooth ice, with  $\mu_{\rm SIT}$  ranging from  $1-1.3\,\mathrm{m}$  and a relatively thin snow layer (SND from  $0.1-0.15\,\mathrm{m}$ ) compared to the MOSAiC Nloop and Sloop. In Fig. 10a, the Log-normal provides the best fit of the observed SND, and the kurtosis values range from 4-13 ( Fig. 11c), higher than those of the MOSAiC Nloop and Sloop. This indicates that the observed SND distribution from the Runway is more sharply peaked and exhibits heavier tails than the Nloop and Sloop, resulting in the best performance of Log-normal.

For the MOSAiC Summer transects measured over SYI between 29 June and 26 July 2020, the snow layer was very thin  $(\mu_{\rm SND} < 0.1\,{\rm m})$  and smooth  $(\sigma_{\rm snow} < 0.07\,{\rm m})$ , while the underlying  $\mu_{\rm SIT}$  exceeds  $2\,{\rm m}$ . In Fig. 10b, the Log-normal consistently provides the best fit. The observed SND distributions are narrowly concentrated and highly asymmetric, with more than 85% transects exhibiting large kurtosis > 10 (Fig. 11d). These results indicate that the Log-normal provides superior performance in representing very thin and smooth snow layers under summer conditions, even when measured over thick SYI.

## **4.2.2** Dependence on ice type and thickness

For all test sites, the fit performance quantified by the mean RMSE values are summarized in Table 2.

**Table 2.** Performance of probability density function (PDF) fitting for snow depth (SND) across all sites. The root-mean-square error (RMSE) values measure the goodness of fitting. The mean sea ice thickness (SIT) and mean SND are calculated. SIT range refers to the difference between the 95th and 5th percentiles of ice thickness for all transects within the site. The best performance PDF (i.e. smallest RMSE value) is in **bold**.

| Test sites      | Ice type | Time         | mean SIT |               | mean    | RMSE (m) |            |       |        |
|-----------------|----------|--------------|----------|---------------|---------|----------|------------|-------|--------|
|                 |          |              | SIT (m)  | (m) range (m) | SND (m) | Skew     | Log-normal | Gamma | Normal |
| MOSAiC - NLoop  | SYI      | Oct - Jan 30 | 3.06     | 6.79          | 0.23    | 0.016    | 0.012      | 0.018 | 0.039  |
|                 | SYI      | Jan 30 - May | 2.90     | 4.13          | 0.29    | 0.009    | 0.037      | 0.013 | 0.039  |
| Lincoln Sea     | MYI      | Apr          | 2.59     | 3.13          | 0.33    | 0.034    | 0.160      | 0.043 | 0.058  |
| MOSAiC - Summer | SYI      | Jun-Jul      | 2.46     | 2.95          | 0.07    | 0.019    | 0.015      | 0.017 | 0.026  |
| N-ICE2015       | SYI      | Jan-Mar      | 1.58     | 2.47          | 0.50    | 0.032    | 0.082      | 0.048 | 0.037  |
| Weddell Sea     | MYI      | Jun-Aug      | 1.43     | 1.91          | 0.53    | 0.015    | 0.051      | 0.027 | 0.016  |
| MOSAiC - SLoop  | FYI      | Oct - Feb 20 | 1.29     | 1.65          | 0.19    | 0.024    | 0.014      | 0.021 | 0.045  |
|                 | FYI      | Feb 20 - May | 1.79     | 1.83          | 0.27    | 0.019    | 0.037      | 0.018 | 0.054  |
| Resolute Bay    | FYI      | Apr          | 1.20     | -             | 0.09    | 0.009    | 0.006      | 0.008 | 0.018  |
| MOSAiC - Runway | NI       | Jan          | 1.16     | 0.38          | 0.12    | 0.026    | 0.016      | 0.022 | 0.038  |
| Weddell Sea     | FYI      | Jun-Aug      | 0.69     | 0.88          | 0.19    | 0.010    | 0.014      | 0.012 | 0.015  |

For SYI and MYI, the Skew distribution best fits sea ice before 30 January in the MOSAiC N-loop, while the Log-normal performs better for deformed ice ( $\Delta_{SIT} > 6 \,\mathrm{m}$ ) and for thin snow over SYI during the MOSAiC summer (see Section 4.2.1). The fitting performance from other sites in addition to MOSAiC are shown in Fig. 13. For the Lincoln Sea where thick MYI dominates, in general, the Skew distributions provide the best fits of the observed SND. This result is consistent with observations from MOSAiC Nloop after 30 January, where the sea ice exhibits similar SIT level. Specifically, Skew yields the lowest RMSE for Sites 1, 2, 4, 5, 6 and NE, while for Site 3, both Gamma and Skew perform comparably. For Site 7 with  $\mu_{\rm SIT}=1.78\,{\rm m}$ , the Gamma provides the best fit. Sites 3 and 7 exhibit higher variability in SND ( $\sigma_{\rm snow}>0.2\,{\rm m}$ ) but lower mean values (0.2-0.25 m) compared to the other sites. This suggests that for surfaces with rougher but thinner snow cover, the Gamma slightly outperforms the Skew distribution. Site 8 exhibits kurtosis values greater than 2 in Fig. 11h, indicating an asymmetric and sharply peaked distribution. In this case, the Log-normal distribution performs best. For N-ICE2015, SYI was covered by a heavy snow layer as thick as  $0.50\,\mathrm{m}$  in average. However, the  $\mu_{\mathrm{SIT}}$  is  $1.58\,\mathrm{m}$ , similar to the Sloop transect but thinner than in the Lincoln Sea, MOSAiC Nloop, and MOSAiC Summer. Overall, as shown in Table 2, the Skew yields the highest accuracy with the smallest RMSE over N-ICE2015. On 7 February and 14 March, PDFs exhibit greater symmetry, with skewness below 0.1 (Fig. 11f). This explains that Skew and Normal perform equally better, while Log-normal does not provide a proper fit, shown in Fig. 13b. The Weddell Sea floe 517 exhibits  $\mu_{\rm SIT}$  of 1.43 m covered by heavy snow layer  $(\mu_{\rm SND}=0.53\,{\rm m})$ , where the snow and ice conditions are similar to those of the N-ICE2015 samples. The Skew provides the best fit, consistent with the performance observed in the N-ICE2015 transects.

For NI and FYI, Log-normal yields the best fitting performance for most cases. Resolute Bay is covered by first-year landfast sea ice, where the snow and ice conditions are similar to the MOSAiC Runway site. The fitting results shown in Fig. 13c indicate that the Log-normal distribution provides the highest accuracy, with the lowest RMSE values. This result is consistent with the results from MOSAiC Runway samples. We also tested thin, young ice from Churchill. Although no co-located ice thickness measurements were available, field observations indicated similar ice conditions to those at Resolute Bay. The results further confirm that the log-normal distribution performs best (see Supplement). For Weddell Sea samples, FYI floes 503 and 506 have an average SIT of  $0.69\,\mathrm{m}$ , which is the thinnest ice in the entire dataset. Floe 506 exhibits higher skewness and kurtosis values greater than 2 in Fig. 11h, indicating an asymmetric and sharply peaked distribution. In this case, the Log-normal distribution performs best, consistent with results of FYI from MOSAiC Runway and Resolute Bay sites. However, floe 503 shows poor performance for the Log-normal distribution. This can be explained by the near-symmetric shape of the SND histogram, with a small skewness value of only 0.26, which diminishes the advantage of Log-normal's ability to model asymmetry. Another exception is MOSAiC Sloop after 20 February, where the best PDF transits from Log-normal to Skew when  $\mu_{\rm SIT}$  reached above  $\sim 1.5\,\mathrm{m}$ , see detailed explanation in Section 4.2.1. It suggests that in addition to the ice age, SIT is a critical parameter to consider when choosing the most appropriate PDF for describing SND distribution.

Figure 13. Root-mean-square error (RMSE) between the fitted probability density functions (PDFs) and the observed snow depth (SND) values. (a) Lincoln Sea. (b) N-ICE2015. (c) Resolute Bay. (d) Weddell Sea. The mean sea ice thickness (SIT,  $\mu_{\rm SIT}$ ) over each transect is calculated.

In order to further generalize the dependence of fitting performance on SIT, we categorize the RMSE values based on μ<sub>SIT</sub> intervals by mixing samples from MOSAiC Sloop, MOSAiC Runway, Lincoln Sea, N-ICE2015, Resolute Bay, and Weddell Sea sites. Note that the MOSAiC Nloop site is excluded since not only SIT but also ice deformation (quantified by SIT range) and meteorological events (e.g., snow drifting) alter the best-fit PDF. The MOSAiC Summer site is excluded since we interpret that for the very thin (μ<sub>SND</sub> = 0.07 m) and smooth snow layer Log-normal performs best and is independent of SIT. In Fig. 14, for ice thinner than 1.5 m, the Log-normal provides the most accurate fit for SND. As the ice thickens, the RMSE for Log-normal increases significantly, while those for Gamma and Skew decrease, indicating improved fitting accuracy. For μ<sub>SIT</sub> between 1.5 – 2 m, both Gamma and Skew perform well, with a slight advantage for Gamma. When ice exceeds 2 m (but with SIT range below 6 m), Skew gives the best fit. Note that the choice of thresholds at 1.5 m – 2 m is based on the observed transition in the best-fitting distribution from the MOSAiC Sloop site time-series observations.

We calculate the RMSE difference ( $D_{\rm RMSE}$ ) between the Log-normal and Skew distributions for each transect to quantitatively identify the best-fitting PDF, as shown in Fig. 15. The same thresholds used in Fig. 14 are applied here to examine the dependence of SND distribution on  $\mu_{\rm SIT}$  and  $\Delta_{\rm SIT}$ . From Fig. 15(a),  $D_{\rm RMSE}$  values are mostly below zero when  $\mu_{\rm SIT} < 1.5\,\mathrm{m}$ , indicating that the Log-normal distribution performs best for NI, FYI, and thinner MYI. The Skew distribution gradually outperforms the Log-normal distribution when  $\mu_{\rm SIT}$  is between 1.5 and 2m. For thicker MYI with  $\mu_{\rm SIT} > 2\,\mathrm{m}$ , the Skew distribution significantly outperforms the log-normal, with  $D_{\rm RMSE}$  deviating substantially from zero.

As discussed in Section 4.2.1, surface roughness is essential in shaping SND distributions, particularly for SYI and MYI.  $\Delta_{\rm SIT}$  as a proxy for deformation level is strongly linked to PDF fitting performance. Fig. 15(b) illustrates  $D_{\rm RMSE}$  for all SYI and MYI samples with  $\mu_{\rm SIT} > 1.5\,\mathrm{m}$ . The skew-normal distribution performs best for less deformed ice with  $\Delta_{\rm SIT} < 6\,\mathrm{m}$ , whereas for highly deformed ice with  $\Delta_{\rm SIT} > 6\,\mathrm{m}$ , likely associated with ridges and strongly peaked SND distributions, the Log-normal distribution performs best.

In summary, we considered the ice ages, SIT, SIT range, and meteorological conditions impacts on the best PDF to describe the SND observations. The results reveal the following patterns:

- (1) For NI and FYI where SIT  $< 1.5 \,\mathrm{m}$ , the Log-normal distribution provides the best fit.
- (2) For SYI and MYI which are typically thicker ice ( $\mu_{\rm SIT} > 1.5\,{\rm m}$ ) and not very deformed ( $\Delta_{\rm ice} < 6\,{\rm m}$ ), as well as FYI grows thicker than  $1.5\,{\rm m}$  (e.g., MOSAiC Sloop), the Skew is generally the best to describe SND distribution. Specifically, for SIT between  $1.5-2\,{\rm m}$ , both Gamma and Skew distributions perform comparably well, while for ice thicker than  $2\,{\rm m}$ , Skew yields the best fit.
  - (3) For SYI and MYI which are typically thicker ice ( $\mu_{\rm SIT} > 1.5\,\rm m$ ) and deformed ( $\Delta_{\rm ice} > 6\,\rm m$ ), such as the Nloop before January 30, the SND distribution can exhibit strong asymmetry, making the Log-normal distribution the best fitting.
- (4) Strong snow drifting events can reduce the peakedness of the SND distribution, leading to a more evenly redistributed snow layer that is better captured by the Skew distribution.
  - (5) During melting season, where the snow layer is very thin and smooth ( $\mu_{SND} 

Figure 14. The boxplot of RMSE between the fitted PDFs and the snow depth (SND) observations for MOSAiC Sloop, MOSAiC Runway, Lincoln Sea, N-ICE2015, Resolute Bay, and Weddell Sea.  $\mu_{\rm SIT}$  denotes the mean sea ice thickness (SIT) of a sub-kilometer scale transect. For each box, the top and bottom boundaries correspond to the third (Q3) and first (Q1) quartiles, and the whiskers extend to the values within Q3 $\pm$ 1.5 $\times$ (Q3-Q1).

## 4.3 Correlation length of snow depth

The correlation length of SND for each transect at all test sites was computed using the method presented in Section 3.2. Note five transects, including MOSAiC Nloop 27 February, MOSAiC Summer 27 June and 30 August, and Lincoln Sea Site 5 and NE, are excluded as their semi-variograms fail to reach a constant semi-variance value, preventing accurate estimation of the sill and effective range (i.e., correlation length). The correlation length of the transects within each site are plotted as boxplots in Fig. 16.

We observe that thicker ice is associated with a larger mean correlation length, shown in Fig. 16. The average correlation length and  $\mu_{\rm SIT}$  of each site are plotted as blue and red lines, respectively, showing a high correlation of 0.92. In general, the average correlation length for NI and FYI is  $39 \pm 25$  m. For smooth and thin snow-covered SYI in summer, this increases to  $54 \pm 28$  m, while snow-covered SYI and MYI in winter shows the longest correlation length, averaging  $72 \pm 25$  m.

400

405

410

Figure 15. The difference in RMSE between the log-normal and skew-normal distributions. A value below (above) zero indicates that the Log-normal (Skew) distribution performs better.  $\mu_{\rm SIT}$  denotes the mean sea ice thickness (SIT) for each sub-kilometer-scale transect. SIT range is calculated by  $\Delta_{\rm SIT} = 95\% {\rm SIT} - 5\% {\rm SIT}$  over each transect.

For the category of NI and FYI, the correlation lengths vary between  $10-80\,\mathrm{m}$ , shown in Fig. 16. The range is consistent with the observations from undeformed FYI found that the  $10\,\mathrm{m}$  length scale is associated with wind-driven dune formation (Moon et al., 2019). At larger scales  $(30-100\,\mathrm{m})$ , the governing processes may involve interactions between individual dunes, such as their gradual movement, merging, or dispersal over time. Specifically, for the three sites (Weddell Sea, MOSAiC Runway, and Resolute Bay) covered by thin ice with  $\mu_{\rm SIT} 

Figure 16. The relation between snow depth (SND) correlation length and sea ice thickness (SIT).  $\mu_{\rm SIT}$  denote the mean SIT over each transect. For each box, the top and bottom boundaries correspond to the third (Q3) and first (Q1) quartiles, and the whiskers extend to the values within Q3±1.5×(Q3-Q1). The red dash and blue dot lines indicate the mean SIT and mean correlation length for each site, with a correlation of Pearson-r = 0.92.

The correlation between the correlation length and the SIT can be explained by the physical processes that shape the snow distribution. The small correlation lengths likely reflect the spatial scale of individual snow dunes and drifts formed by wind; beyond short distances, the SND at one location provides little information about the depth at another. In contrast, on rougher MYI, larger correlation lengths are expected, as prominent surface features such as hummocks and ridges affect snow accumulation over broader spatial scales.

Within each site, differences in correlation lengths (Fig. 16) can be explained by the inherent spatial variability of snow distribution such as the anisotropic structure of snow dunes (Iacozza and Barber, 2010), as well as changes to these patterns temporally due to the precipitation and wind events (Itkin et al., 2023). To further investigate the temporal changes of the spatial heterogeneity of snow cover, we analyzed time-series variograms from the MOSAiC Sloop, Nloop, Runway, and Summer transects. For the MOSAiC Nloop site, we observe a significant increase in the correlation length following the snowfall and snow drift events from 30 January to 3 March and from 20 February to 20 March (Fig. 17a). A major snow drift event occurred on 24-25 February, coinciding with the largest correlation length observed on 27 February, indicating a snow correlation length exceeding 150 m. Slight increases in snow correlation length were also observed on 5 December, 24 April, and 7 May, coinciding with snowfall events. The correlation length increases during these periods, suggesting growth in snowdrift size. One

possible process is that during significant snowfall or drift events, the valleys between the snow dunes become filled, leading to larger and more continuous snow structures, as revealed by geostatistical analysis (Iacozza and Barber, 2010). This infill significantly alters the pattern of the SND distribution and influences subsequent snow redistribution processes. For MOSAiC Sloop, although we do not observe changes in snow correlation length as pronounced as those at MOSAiC Nloop, there is a steady increase from around 28 m to 60 m during the drifting event from 20 February to 20 March. In addition, small increases in correlation length were noted after snowfall events on 5 December, 30 January, and 26 April. These results suggest that drifting snow events have a greater impact on increasing snow correlation length over rougher ice (MOSAiC Nloop) compared to smoother ice (MOSAiC Sloop).

**Figure 17.** The time-series correlation lengths over MOSAiC (a) Nloop and (b) Sloop. The relevant climate events such as storm, snowfall, and snow drift are from (Wagner et al., 2022).

## 5 Conclusions

We presented a detailed examination of sub-kilometer-scale SND distributions as a functions of SIT across various sea ice types and seasonal conditions. The dataset includes field observations from MOSAiC (Nloop, Sloop, Runway, Summer transects), the Lincoln Sea, N-ICE2015, Resolute Bay, and the Weddell Sea, spanning NI, growing and smooth FYI, flooded ice, rough SYI, thin-snow MYI in summer, and MYI in winter.

First, we characterized the statistical properties such as mean and variability of SND and its dependence on SIT. We found that (i) for NI and FYI ice, SND increases linearly with SIT. This reflects that younger ice forms concurrently with snow cover during the same seasonal cycle, preserving a linear relationship between SND and SIT. (ii) CV remains independent of SIT and type with a consistent coefficient of variation (CV  $\approx 0.50$ ), allowing estimation of the SND variability from the mean SND values. (iii) Lower CV for the flooded site is the result of a lower than expected SND variability. The phenomenon of flooded sites diverging from the CV of non-flooded sites can be a call for further research, both in the field but also in the lab and with models.

Second, we fitted the SND distribution with Normal, Log-normal, Gamma, and Skew distributions and quantified the goodness of fit using RMSE values from QQ-plot. We observed a dependence of the fitting performance on ice ages, SIT, deformation (measured by the SIT range), and meteorological conditions: (i) For NI and FYI where SIT <  $1.5\,\mathrm{m}$ , the Log-normal best describes the SND distribution. (ii) For SYI and MYI, as well as thick FYI (SIT >  $1.5\,\mathrm{m}$ ), Skew is generally superior, especially when SIT >  $2\,\mathrm{m}$ . (iii) For SYI and MYI which are thick and heavily deformed ice (SIT >  $1.5\,\mathrm{m}$  and SIT range >  $6\,\mathrm{m}$ ), the Log-normal distribution again provides the best fit due to the strong asymmetry in the SND distribution, highlighting ice deformation as a key factor in determining the most appropriate statistical model. (iv) Strong snow drift can flatten the SND distribution (decrease the kurtosis value), making the Skew distribution more suitable for representation. (v) During melting season (thin and smooth snow layers), Log-normal fits best even under thick ice.

Finally, we characterized the snow correlation length using semi-variogram analysis and found that (i) thicker ice is generally associated with larger snow correlation lengths. For NI and FYI, the average correlation length is  $39 \pm 25$ m. For smooth and thin snow-covered SYI in summer, it increases to  $54 \pm 28$ m, while snow-covered SYI and MYI in winter exhibits the longest correlation length, averaging  $72 \pm 25$ m. The observed positive relation between the correlation length and SIT can be attributed to the physical processes shaping snow distribution. Shorter correlation lengths likely reflect the spatial scale of individual snow dunes, where the snow layer varies rapidly over short distances. In contrast, rough SYI and MYI with prominent surface features such as hummocks and ridges promote snow accumulation over broader spatial scales, resulting in larger correlation lengths. (ii) Furthermore, snowfall and drift events were found to enhance the correlation length, particularly in SYI than in FYI, by filling the valley between the snow dunes and strengthening large-scale snow features.

These findings emphasize the need for SIT-dependent parameterizations in climate models and remote sensing applications, particularly for accurately capturing SND variability at sub-grid scales. Future work involves further characterizing the SND regional and temporal variation by integrating high-resolution surface topographic measurements (e.g. terrestrial laser scanner) of snow bedforms and incorporating the predicted SND distribution with snow models.

## 470 Code availability

. The code needed to replicate this analysis will be publicly available via GitHub after the manuscript being accepted.

## Data availability

. MOSAiC data can be accessed from the PANGAEA website (https://doi.pangaea.de/10.1594/PANGAEA.937781, Itkin et al. 2021). The N-ICE2015 data can be accessed from the Norwegian Polar DataCentre (https://data.npolar.no/dataset). The Weddell Sea data can be accessed from the PANGAEA website (https://doi.org/10.1594/PANGAEA.933584). The Resolute Bay data and Lincoln Sea data can be accessed from personal contact to Prof. Julienne Stroeve.

#### **Author contributions**

. LH and JS were involved in the conceptualization of the study. LH, JS, TN, RM, RW, LZ, and MJ were involved in developing methodology and analyzing results. JS, RW, CN, AF were involved in planning of the Resolute Bay field campaign and collected the snow and sea ice data from the field. LH prepared the original draft. All co-authors were involved in the review and editing process.

#### **Competing interests**

. The contact author has declared that none of the authors has any competing interests.

### Acknowledgements

. The authors thank all individuals involved in the field campaigns, including AWECS (2013), N-ICE2015, CryoVEx-2017, MOSAiC (2019–2020), and Resolute Bay (2025). We are especially grateful to Prof. Christian Haas for providing co-located snow and ice data from the Lincoln Sea 2017 campaign. Datasets used in this paper were produced as part of the MOSAiC, with the tag MOSAiC20192020 and the project ID AWI\_PS122\_00. We thank all people involved in the expedition of the Research Vessel Polarstern ((für Polar-und Meeresforschung, 2017)) during MOSAiC in 2019–2020, as listed in (Nixdorf et al., 2021). Lanqing Huang received financial support for this research from a Swiss National Science Foundation Postdoc.Mobility Fellowship (grant no.P500PN\_217817). Carmen Nab acknowledges support from the Horizon 2020 CRiceS grant (#101003826). Alicia Fallows was supported by the Natural Environment Research Council (grant number NE/S007229/1) and the NERC DEFIANT Grant (NE/W004712/1) through JS and RW.

# Appendix A: The statistics of SND and SIT over the study area

The distribution of SND and SIT on each transect is calculated and plotted in Fig. A1 and Fig. A2.

# Appendix B: The geolocation of the MOSAiC loops

The Geolocation of the time-series measurement from MOSAiC Nloop, Sloop, Runway, and Summer loops are shown in Fig. B1.

# **Appendix C: PDF fitting performances**

The histograms of the observed SND and the fitted PFDs from Skew, Gamma, Log-normal, and Norm distributions are shown in Fig. C1.

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

**Figure A1.** Violin plots of the snow depth (SND) over each transect from campaign fields. The red dot and black line represent the mean and median values, respectively.

**Figure A2.** Violin plots of the sea ice thickness (SIT) over each transect from campaign fields. The red dot and black line represent the mean and median values, respectively.

**Figure B1.** The geolocated snow and ice thickness measurements for (a) Nloop, Sloop, and the Runway, and (b) the Summer loops during MOSAiC. (c) A zoom in of comparing the geolocations for 16 January (red) and 2 February (black) transects. The basemaps depict sea ice topography, represented by total freeboard measurements derived from Airborne Laser Scanning on (a) and (c) January 28 and (b) May 10, 2020, respectively.

Figure C1. Display of the histograms and the fitted probability density functions (PDFs) over selected transects as examples.