# Peer review of "Sub-kilometer Scale Snow Depth Distribution on Sea Ice of Different Ages and Thickness"

_EGUsphere, 2025_

## Community Comment (CC1)

Hey all,

I think there lies great potential in the deployed methodology to evaluate snow distributions and their statistical properties. I appreciate the effort the authors have put in the geostatistical analysis connecting snow depth and sea-ice thickness measurements. With this comment I want to raise some points regarding the "correlation length scale" methodology and results:

During recent works on the same topic and similar data, I found the results and performance of semi-variograms to be significantly impacted by input parameters, such as (a) the chosen lag bin widths (and whether they are constant), (b) the careful filtering of data and (c) the choice of model to fit the empirical variogram.
I.e., for the example data *(MOSAiC Sloop on the 14th of November 2019)* which you show in your Figure 6, there is one outlier snow depth data point (see Figure 1 below). I am not very sure whether it is a wrong measurement or an actual snow depth value. However, it impacts the derived semi-variogram majorly (see Figure 2). Did you perform any filtering on the data sets?

[Figure]

Figure 1: Left: FloeNavi map of the Magnaprobe snow depth values in the Sloop on the 14th of November 2019. Right: Histogram of the same snow depths.

[Figure]

Figure 2: Left: Unfiltered semi-variogram. Right: Semi-variogram after filtering the outlier snow depth data point.

You can see, that the retrieved variogram and thus the correlation length scales vary greatly between these two data sets – even though it is only one point that is different. In the Figure 2, I have used two models to fit to the experimental variogram, the Matern model (see Equation below) and the exponential model which you use as well.

$$\gamma_z(h) = c_0 \left[ 1 - \frac{2}{\Gamma(\nu)} \left( \frac{h\sqrt{\nu}}{a_0} \right)^\nu K_\nu \left( 2\frac{h\sqrt{\nu}}{a_0} \right) \right]$$

Where $h$ is the lag bin, $\nu$ is a smoothing factor (for $\nu = 0.5$, the Matern model is equal to the exponential model, for $\nu > 10$ it approaches a Gaussian model). $c_0$ is a rescale factor (usually $c_0 = 1$), T is the Gamma function and $K_\nu$ is the modified Bessel function of the second kind. $a_0$ is the range or correlation length. What was your choice to use only an exponential model for the fit? Also, how did you choose the lag bin width of 3 m?

For the extensive ECCC Eureka 2014 and 2016 data sets published by Josh King (https://github.com/kingjml/ECCC-Eureka-2014-Snow-on-Sea-Ice-Campaign and https://github.com/kingjml/ECCC-Eureka-2016-OpenData. Maybe you want to include that data as well? It has both MP and EM31 SIT data!), I have found the Matern model with $\nu = 1$ fits the experimental variograms best as it can resemble both the "flattening" at short length scales (see the example in Figure 3) but also a sharp decrease as seen in Figure 2, right panel.

[Figure]

*Figure 3: Semi-variogram of Grid8 from the ECCC 2016 Eureka snow on sea ice campaign. The "flattening" at shorter length scales is well covered by the Matern model. Figure adapted from Kagel (2025).*

As this method is to my knowledge not commonly used in snow/ sea-ice sciences, it could be valuable to include a more sophisticated assessment of the derived geophysical information (i.e., correlation lengths and dependence of ice type and SIT) by motivating choices of parameters and showing the robustness by using different models and/ or e.g., a Monte-Carlo approach to sub-sample the individual datasets. I think this can also prove the statistical significance of the time series you show in Figure 17. At the current stage, I don't think the shown methodology is robust enough to prove that this is an actual signal and not random due to i.e., better or worse fits.

Best, Torbjörn Kagel